# Keeping People with Dementia or Mild Cognitive Impairment in Employment: A Literature Review on Its Determinants

**DOI:** 10.3390/ijerph17030842

**Published:** 2020-01-29

**Authors:** Fabiola Silvaggi, Matilde Leonardi, Pietro Tiraboschi, Cristina Muscio, Claudia Toppo, Alberto Raggi

**Affiliations:** 1UOC Neurologia, Salute Pubblica e Disabilità, Fondazione Irccs Istituto Neurologico Carlo Besta, 20133 Milan, Italy; matilde.leonardi@istituto-besta.it (M.L.); claudia.toppo@istituto-besta.it (C.T.); alberto.raggi@istituto-besta.it (A.R.); 2UOC Neurologia 5 – Neuropatologia, Fondazione Irccs Istituto Neurologico Carlo Besta, 20133 Milan, Italy; pietro.tiraboschi@istituto-besta.it (P.T.); cristina.muscio@istituto-besta.it (C.M.)

**Keywords:** employment, dementia, work performance, work engagement, public health

## Abstract

*Background:* Approximately 10–20% of people with early onset dementias (EOD) or mild cognitive impairment (MCI) are aged under 65 and, due to extended working life and increasing prevalence of dementias, they more and more frequently will be present in the active workforce. This review aimed to synthesize the available information about the ability of people with EOD or MCI to retain their participation in the labor workforce. *Methods*: We searched SCOPUS and EMBASE for peer-reviewed papers that reported studies assessing work ability in employees with EOD or MCI that were published in the period of January 2010 to August 2019. *Results*: We selected four publications, in which 1012 participants with EOD or MCI were enrolled (41.2% males). Cognitive difficulties rather than motor dysfunction were found to reduce patients’ ability to work. Two main themes emerged: management of dementia in the workplace and the impact of symptoms on working status. *Conclusions*: EOD and MCI impact on workforce participation by determining problems in executive functions. Although this review was based on a small sample of studies, it can be shown that support in the workplace may act as a facilitator to enhance workforce participation, and occupational health professionals can help patients with EOD or MCI continue working as much as possible.

## 1. Introduction

Dementia is a broad term referring to a number of neurological disorders that cause a progressive decline in cognitive functioning [1] and can be associated with several neurological conditions, such as Parkinson’s disease (PD), Huntington’s disease (HD), Alzheimer’s disease (AD) and AD-related disorder (ADRD), Lewy bodies dementia (LBD), frontotemporal dementia (FTD), or vascular conditions [2].

The vast majority of dementia cases are diagnosed among people aged 70 or older, and few are diagnosed in adulthood, that is, before the age of 65—these conditions are referred to as early-onset dementia (EOD). EOD incidence has been estimated at 13.4/100,000 persons per year in the 30–64 year old age group [3], while its prevalence in the last 15 years was reported between 42.3 and 81.0 per 100,000 cases [4,5,6].

In addition to this, dementia can be preceded by mild cognitive impairment (MCI), whose prevalence is 3–22% in people aged 65 or older and can also be diagnosed before aged 65 in a portion of the population wherein it progresses into dementia in 5–10% of cases per year [7]. 

Thus, there is a portion of working people that may experience difficulties with work-related activities, or have a reduced work ability, due to EOD or MCI; worldwide, it is in fact estimated that 10% of the 35.6 million people living with dementia are aged under 65 [8].

Some workers with EOD or MCI continue to work, thanks to the adjustments to their activities made by companies, whereas others have to quit because of impaired performance [9,10]. 

Symptoms of cognitive decline are in fact frequently first noticed by caregivers or relevant others—employed people, co-workers, clients, and employers may notice the presence of cognitive difficulties because they interfere with an individual’s ability to work, expressed as being easily distracted by background noise, poor memory, and inability to perform multi-tasking activities [11].

It has to be noted that these symptoms also affect older adults without dementia and may cause a reduction in their work performance. Older workers are often considered less flexible, with a lower ability to learn and with more difficulty to train, due to a normal decline of cognitive ability and motor functioning [12,13]. Organizational practices often dissuade older employees from participating in training [14,15], with implications being on the assignment of job tasks. Older workers are not always deemed suitable for tasks that are commonly associated with younger workers, and may cause a reduction of self-esteem and motivation, as well as difficulty in remaining in the labor market [16].

Dementias and MCI are expected to increase in prevalence due to population ageing [17], early diagnosis due to the increasing use of biomarkers [18,19], as well as due to increased risk of higher prevalence of conditions that increase the risk of dementia such as type-2 diabetes mellitus [20,21].

Parallel to this, working life will be extended in the near future due the ageing population, less replacement from younger workers, and national welfare policies encouraging older workers to remain in the labor workforce, delaying retirement [10,22]. The joint effect of these three phenomena will likely determine a higher presence in the workplace of people with cognitive impairments, MCI, or EOD.

It is important in any workplace to consider and decide what kind of interventions could lead to better work performance for people with brain disorders associated with cognitive deterioration [23]. Work-site interventions are feasible if they take into account the balance between the job demands on one side and, on the other side, the mental and physical resources of workers that, in this case, are impaired as a consequence of cognitive difficulties [24].

The relationship between job demands and workers’ resources is known as “work ability” [22], and is determined by professional knowledge and skills, values, attitudes, motivation, and features of work itself [23]. Helping workers to maintain their work ability is beneficial to companies, as it has the potential to enhance workers’ productivity and strengthen their mental and physical resources [25,26].

To keep on working has several advantages for people with cognitive impairments, MCI, or EOD. Beyond economic benefits, social and psychological benefits, including self-esteem, personal identity, quality of life, and full participation in society are of importance [27].

Many studies, which have focused on the problems associated with an unplanned early labor market exit, have evidenced several negative consequences, including increased incidence of depression [28], loss of personal identity [29], and loss of social networks [30].

For this reason, it is important that the workers collaborate with Human Resourcesstaff in planning interventions aimed at fostering their work ability [31]. In planning workplace support activities, they should consider the severity of workers’ cognitive impairment and manage its related stigma through clear and inclusive praxes in which colleagues are informed on the potential benefits of supporting people who develop cognitive impairment [32].

In sum, EOD and MCI are conditions whose typical age of onset is around the period of retirement, and little research has addressed the work-related experience of people with this symptomatology [21,33], as well as the social and economic consequences of their exit from the labor force [34,35,36].

For this reason, understanding the capacity of people with EOD and MCI to keep on working, that is, understanding their work ability, as well as understanding what can be done in the workplace and how to deal with factors associated with job loss, will be of importance for public health policies.

As such a piece of information has not yet been synthesized in a literature review, the aim of the present study was to provide updated information about the ability of people with EOD and MCI to keep on participating in the labor workforce.

## 2. Materials and Methods

### 2.1. Search Strategy

We performed a comprehensive search on SCOPUS and EMBASE, covering the period of January 2010 to August 2019 to identify primary research papers reporting either randomized clinical trials (RCTs) or observational studies that assessed work ability in workers with EOD or MCI, or the factors associated with job loss.

The following combinations of keywords were searched within the titles, abstracts, or keywords: (dementia OR “mild cognitive impairment” OR MCI) AND (“work performance” OR “work ability” OR “performance appraisal” OR employment).

Our search was limited to original studies, published in English and with an abstract, which had to be indexed by SCOPUS or EMBASE. Please see the Appendix A for the detailed search strategy.

### 2.2. Articles’ Inclusion and Exclusion Criteria

We specifically searched for clinical trials and observational studies, either cross-sectional or longitudinal, and excluded reviews, commentaries, letters to the editors, editorials, qualitative studies, and case reports.

The content of papers had to enable the extraction of information on work ability, defined as work performance or performance appraisal, in workers with EOD associated with different neurological conditions (i.e., PD, HD, AD, ADRD, LBD, FTD, or vascular conditions) or MCI. 

Studies drawing from populations wherein the presence of EOD or cognitive difficulties was chiefly addressed as a symptom in the context of other general medical conditions were excluded. Finally, studies that were based upon caregivers’ reports only were excluded as well.

### 2.3. Paper Selection and Data Extraction

Abstracts of papers were screened by a junior researcher (F.S.). To ensure quality and consistency of data extraction, 20% of the abstracts and of full texts were randomly selected for a second check by a senior researcher (A.R.) who was blind to the decision of the firstcheck. We determined the overall agreement rate between researchers—if it was below 70%, each of the double-checked abstract or manuscript was re-reviewed again by the two researchers to get to a final decision by consensus, and an additional 20% set of abstracts and full texts was double-checked again.

Extracted information included health conditions—broadly defined as unspecified EOD; EOD due to a specific underlying condition, i.e., PD, HD, AD, ADRD, LBD, FTD, vascular conditions; MCI—and the main characteristics of patients, which included sample size, gender distribution, age, and percentage of employed subjects for each study.

As we had foreseen that little literature was available on this topic, we did not organize the main contents on work ability into overarching categories. Rather, available information on work ability was descriptively reported and we relied on a bottom-up approach for the organization of review’s results, that is, we described literature findings consistently with the way in which results were reported in each paper. 

Whether any issue associated with or determinant of work ability was found, the content of the manuscript was included under a topic, and the same procedure was followed in case the same topic was addressed in other studies.

## 3. Results

The initial search returned 1675 records. Following abstract screening and full text assessment, four publications were selected for inclusion in this review [37,38,39,40]. The rate of agreement between reviewers was 99.8% at the abstract check and 100% at full-text evaluation. 

Figure 1 shows the flow diagram of our search process and Table 1 presents summary findings of the publications included in this review.

Across the studies, 1012 participants with a diagnosis of neurological conditions associated with dementia were enrolled, with men being a minor part (41.2% of the entire group)—two studies involved samples of patients with unspecified EOD, and two with EOD due to HD. 

One study addressed the management of patients in the workplace [37], and three prospectively addressed working status change—one addressed job loss after unspecified EOD diagnosis [38], and the second and third evaluated the negative impact of neuropsychiatric symptoms in patients with EOD due to HD [39,40].

### 3.1. Management of Dementia in the Workplace

One study addressed the management of patients with unspecified EOD in the workplace [37]. The paper described a demonstration program called Side by Side to evaluate the possibility of engaging patients with EOD through work-site interventions. The program started in August 2011 and was still ongoing when the paper was published. 

The program involved seven people with mild EOD aged between 50 and 65 years. They worked one day per week in a hardware store with the support provided by work-buddies and program staff. The participants themselves chose the area of the store where they wished to work and undertook the work duties related to that specific area. 

The work day started with an appointment in a cottage of Life Care’s community, where all participants were subjected to safety checks and briefing and then travelled to their workplace in a minibus.

In a hardware store, the participants worked for a 4 hour shift with a mid-shift meal break.

End of day activities included debriefing to identify and discuss any difficulties encountered during the work shift, journaling to provide a monitoring of the work experience for each participant, and socialization with the group. 

All participants were able to work and enjoyed their workplace experience, showing interest and an improvement of self-esteem, as reported by family carers.

### 3.2. The Impact of Symptoms on Working Status

Three studies addressed working status change. Sakata and Okumura conducted a matched cohort study to show the incidence of job loss among patients with unspecified EOD [38].

The study involved 220 participants with EOD aged 40–59, and 1100 age and gender-matched controls without EOD. The data showed that by 2 and a half months after EOD diagnosis, there was an increased likelihood in leaving one’s job. After 6 months, 9.1% of participants with EOD and 3% of non-EOD controls left their job. After 1 year, the same figures were 14% and 7.3%, respectively. 

The second study evaluated which HD-related signs are associated with unemployment [39] in patients with EOD due to HD. The study involved 220 HD mutation carriers that were categorized as employed (*n* = 114) and unemployed (*n* = 106). In the multivariate analysis, the signs that were significantly related to unemployment state were apathy, cognitive flexibility, and executive functioning domains, with apathy being the strongest sign related to unemployment. 

The third study evaluated the association of cognitive decline, compared with motor decline, with the decision to leave work [40] in patients with EOD due to HD. The study involved 642 patients with HD aged 18–64, recruited from the Enroll-HD observational study of 2015. Results showed that every year of delay in cognitive symptom onset corresponded to approximately 0.8 years of delay in retirement age (approximately 0.5 after correcting for confounders). 

Parallel to this, every year of delay in motor symptom onset corresponded to approximately 0.8 years of delay in retirement age (approximately 0.4 after correcting for confounders). 

Moreover, post hoc analysis revealed that presence of both motor and cognitive symptoms was associated to a higher likelihood of quitting work.

## 4. Discussion

With this review, we aimed to identify and synthesize the available knowledge regarding the ability of people with EOD and MCI to remain in the labor workforce, as well as the factors associated with job loss. The results strengthen the idea that patients with EOD have an increased likelihood to leave their job due to the impact of symptoms on the ability to work, as well as due to the workplace’s inability to adapt to the person’s needs. The symptoms that have been reported as reducing patients’ ability to work are mostly related to cognitive and behavioral deficits, such as apathy and cognitive impairment, rather than to motor dysfunction. One study reported the results of a work participation program dedicated to patients with EOD that, although based on a very limited number of participants, showed some positive results. Finally, our study points out the limited attention that is given to such a topic so far; in fact, our results are based on a very small amount of manuscripts, which represent approximately 0.2% of all the retrieved records.

It has to be noted that the participants of the studies herein included were mostly women of an older age, which basically corresponds to the typical presentation of dementia. A previous report has moreover shown that women with dementia have a more rapid decline cognitive compared to men [41]. This situation impacts their participation in the workforce. Our review highlights the fact that women were more likely than men to leave their job within 1 year after their diagnosis. This evidence is similar to the results of prior studies examining the increased risk of job loss associated with health problems [42].

Apathy is a commonly reported symptom in patients with dementias and MCI [39], and it can have a major impact on health-related quality of life [41]. It is connected to a decline in general functioning defined as loss of autonomy in performing activities of daily living and executive dysfunction [43], and it has been found to be related to unemployment in people with EOD due to HD [37]. However, apathy can be difficult to measure as a consequence of the absence of a clear definition—where different concepts such as lack of interest, abulia, affective flattening, social withdrawal, and indifference coexist [44]—and thus can be misinterpreted in workplace contexts and taken as laziness.

Conversely, depression was not associated to unemployment status [38]. Such a results is at least unexpected because different studies, including clinical studies and population surveys [45,46] found the opposite association. It should moreover to be noted that in the sample of the study by Robertson [37], the overall level of depression was moderate to severe, with half of the participants referring depressive symptoms. Cognitive difficulties, such as visual attention, processing speed, and cognitive flexibility, i.e., executive functions, have been found to be associated with unemployment [38].

Executive functions include planning, organization, cognitive flexibility, and behavior regulation. Thus, patients with executive function deficits may fail in understanding what is needed to complete a task and may be unable to focus their attention on more than one thing at a time.

The association between deficits in executive functions and unemployment is therefore expectable, as these functions constitute the soft skills, that is, the cluster of personality traits and personal habits, including work ethic, courtesy, teamwork, self-discipline, self-confidence, conformity to prevailing norms, and language proficiency [47], that are needed to comply with the vast majority of job tasks. Previous researches have highlighted the importance of soft skills in workplace contexts; in fact, it has been demonstrated that 75%–85% of long-term job success depends on people’s ability to utilize soft skills, and only 15%–25% is determined on technical knowledge [48,49].

Soft skills are considered extremely important in many occupations and industries [50,51], and impairment of these skills due to emerging cognitive difficulties makes it difficult for dementia workers to continue to work. This is confirmed by Worach-Kardas and Kostrzewski, who found that any impairment in mental health can result in unemployment [52].

The decision or need to leave work is influenced by the onset and presence of both cognitive or motor symptoms, and such a concept was addressed in one study in which patients with EOD due to HD were enrolled [39].

A previous study has revealed that the participants with higher baseline cognitive scores had slower decline in functional capacity, demonstrating a marked decline of both cognitive symptoms and motor symptoms [53]. It is, however, difficult to distinguish the impact of motor vs. cognitive symptoms in patients with HD, as they are closely interlinked—this is consistent with the idea of cortical loops subserving the coordination of motor, cognitive, and emotional function, integrated in the basal ganglia [54].

On the other hand, some patients, in the mild stage of dementia, retain the ability to work, provided the adjustments to their activities made by employers, as shown in a recent qualitative report [55]. This is consistent with work-site interventions that provide an opportunity for people with EOD to undertake tailored activities, with the constant presence of staff supervisors [37,56,57,58,59].

Taking into consideration the trends towards increasing working age [19,20] coupled with increase in the prevalence of neurological conditions associated with dementia and cognitive decline, also due to early diagnostic ability [12,13,14,15,16], it is reasonable to expect a rise in the presence of people with EOD or MCI in the labor force. Such a phenomenon has implications both for employees and employers, as well as for those working independently.

By keeping on working, people may gain in overall functioning and quality of life and preserve positive health outcomes and wellbeing, whereas the latter may benefit from avoiding immediate loss of skilled personnel and may enhance the general workplace environment and the companies’ social responsibility, for example, in helping to reducing stigma and promoting social inclusion [60].

However, there are intrinsic difficulties that deal with the impairments in both technical and soft skills due to dementias or MCI, which call for counselling and professional support in the workplace for workers and for managers.

It is our opinion that such services should be routinely provided to workers with EOD and MCI to extend their professional lives as much as possible.

Occupational health professionals can help patients continue working [30]. This, besides the aforementioned positive effects for workers and employers, may in turn impact on the indirect costs of dementias, that is, those connected to premature retirement, which account for approximately 36% to 60% of total dementia cost [60].

To maximize the outcome in terms of preserved work ability, it is our opinion that support services for employment should be integrated within the care pathway, and that legal disposition encourage older workers to remain in the labor workforce to the most prolonged period possible.

One limitation of this review needs be acknowledged. Even though our search was quite extensive, we cannot be sure that all relevant articles were found. In addition to this—although this is not a proper limitation—an overall caution is needed with the interpretation of our results in consideration of the fact that only 0.2% of retrieved records were finally included in our review.

## 5. Conclusions

This review points out, as a first result, the paucity of literature addressing the work ability and factors associated with job loss in people with EOD, along with the absence of studies addressing the same issues in people with MCI.

Such a fact calls for action, as it is likely that in coming years more and more people with such conditions will need information on how these symptoms should be handled in the workplace, as well as patients’ associations needing evidence to develop recommendations for the working sector so as to avoid premature leaving of jobs of patients with mild symptoms of dementia.

Keeping in mind the small amount of studies, we can cautiously conclude that EOD, associated with HD or of an unspecified origin, impacts upon workforce participation, mostly due to neuropsychiatric symptoms that may determine problems with the so-called “soft skills”.

However, one experience that was found was that support in the workplace may act as a facilitator in enabling patients’ ability to carry out daily work tasks.

Such work-site interventions may contribute to preserve patients’ overall health outcomes and wellbeing, reduce stigma, promote participation and social inclusion, and, in turn, might impact on societal costs of dementias through a containment of indirect costs.

Future studies should take these aspects into account and investigate which signs and symptoms are mostly connected to job loss across different kind of EOD and MCI, so that strategies aimed at enhancing work retention in these patients can be planned.

## Figures and Tables

**Figure 1 ijerph-17-00842-f001:**
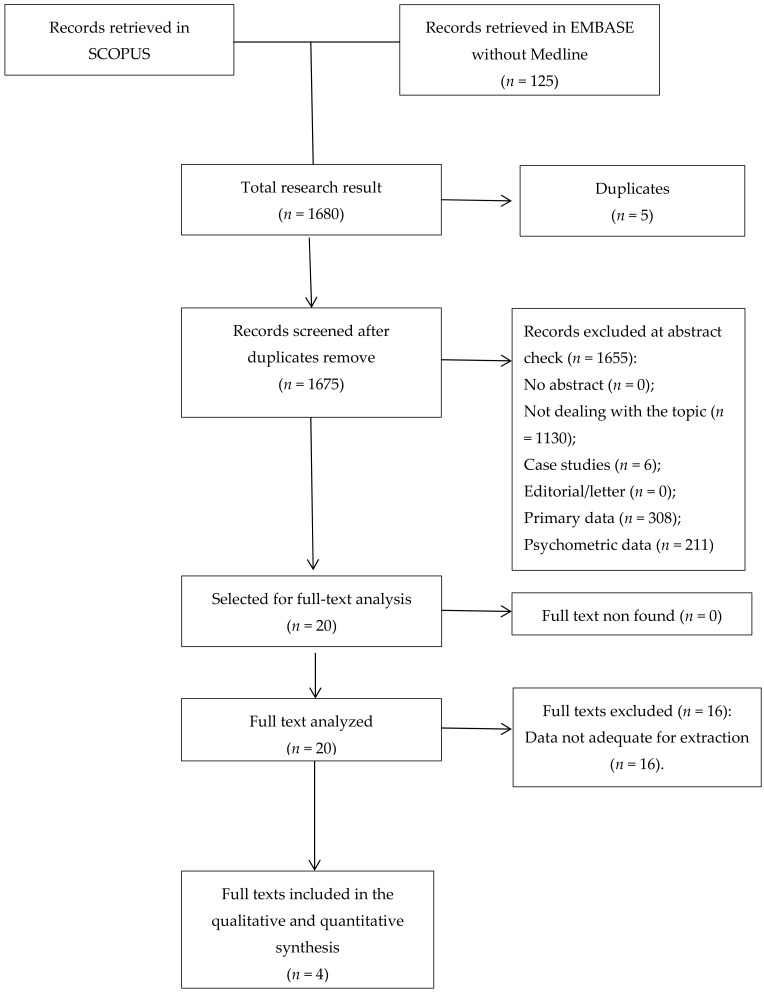
Flowchart of papers’ selection.

**Table 1 ijerph-17-00842-t001:** Main characteristics and main outcomes of the included studies.

ID	Dementia Type	Sample Size	Age	Male Gender (%)	Employed (%)	Main Results
Robertson et al., 2013 [37]	EOD	7	57.5 (50–65) ^b^	4 (57.1%)	7 (100%)	Appropriate support povided in the workplace is associated with improvement of the work ability in patients with EOD.
Jacobs et al., 2018 [38]	HD	220	46.6 (19–75) ^b^	96 (43.6%)	114 (52%)	Apathy and cognitive impairments (information processing speed and cognitive flexibility) are determinants of unemployment in patients with HD.
Sakata et al., 2017 [39]	EOD	143	53 (48–57) ^a^	19 (13%)	143 (100%)	A total of 14% of patients with EOD left their jobs within one year after their diagnosis (7.3% among those without EOD).
Watkins et al., 2018 [40]	HD	642	18–65	298 (46%)	305 (44%)	Cognitive symptoms and motor symptoms have a significant influence on decline of work ability and on the decision to leave work in patients with HD

Notes: ^a^ median and interquartile range; ^b^ age range. EOD: early onset dementia, HD: Huntington’s disease.

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
