# Peer review of "Keeping People with Dementia or Mild Cognitive Impairment in Employment: A Literature Review on Its Determinants"

_ijerph, 2020, doi:10.3390/ijerph17030842_

Round 1

Reviewer 1 Report

This paper tries to address an important topic. Interesting themes were discussed but I find the paper hard to follow. There were too many paragraphs with one to two sentences, resulting in somewhat disjointed arguments. Would the authors consider rewriting better paragraphs.

The second point concerns the context. Employment issues are somewhat tricky even for older adults in general (i.e. without dementia etc). The authors should consider reviewing some core papers in this area to present a richer and more comprehensive background to the work. Issues like cognitive decline and motor functioning are also factors hindering the employment of older adults in general..

Author Response

To the Editor

International Journal of Environmental

Research and Public Health

Title of paper: Keeping people with dementia or mild cognitive impairment in employment: a literature review on its determinants

Corresponding author: Fabiola Silvaggi, PhD

Address of corresponding author:

UOC Neurologia, Salute Pubblica e Disabilità

FONDAZIONE IRCCS ISTITUTO NEUROLOGICO CARLO BESTA

Via Celoria 11, 20133 Milano (Italy)

Telephone: +39.02.2394.3105

Email: fabiola.silvaggi@istituto-besta.it

Dear Editor and reviewers,

We were pleased to receive your response letter together with a set of useful comments on our manuscript entitled “Keeping people with dementia or mild cognitive impairment in employment: a literature review on its determinants”, for which we really thank you.

We decided to undertake the revisions and tried to respond to each of the reviewers in the most appropriate way. Changes in the manuscript are highlighted in yellow, so that changes are easily visible to the editors and reviewers.  Responses to reviewers are also reported in the present letter and in the specific form on the submission system.

Thank you for your attention, best regards

Fabiola Silvaggi on behalf of co-authors, Milano – 11/01/2020

Reviewer 1

Comments and Suggestions for Authors

This paper tries to address an important topic. Interesting themes were discussed but I find the paper hard to follow. There were too many paragraphs with one to two sentences, resulting in somewhat disjointed arguments. Would the authors consider rewriting better paragraphs.

RESPONSE 1: Thank you for this comment, we totally agree and we reviewed some paragraphs.

The second point concerns the context. Employment issues are somewhat tricky even for older adults in general (i.e. without dementia etc). The authors should consider reviewing some core papers in this area to present a richer and more comprehensive background to the work. Issues like cognitive decline and motor functioning are also factors hindering the employment of older adults in general.

RESPONSE 2: We thank reviewer for this very interesting consideration. We have included this information in introduction (lines 52-58).

Reviewer 2 Report

The author provides a comprehensive review including the effects of EOD and MCI on workforce participation.

Following changes need to be addressed:

Comparison of the effects of EOD and MCI on workplace performance between males and females should be discussed in detail. Demographic differences should also be accounted for. 'screened' is misspelled in figure 1. data from other diverse sources should be included.

Author Response

To the Editor

International Journal of Environmental

Research and Public Health

Title of paper: Keeping people with dementia or mild cognitive impairment in employment: a literature review on its determinants

Corresponding author: Fabiola Silvaggi, PhD

Address of corresponding author:

UOC Neurologia, Salute Pubblica e Disabilità

FONDAZIONE IRCCS ISTITUTO NEUROLOGICO CARLO BESTA

Via Celoria 11, 20133 Milano (Italy)

Telephone: +39.02.2394.3105

Email: fabiola.silvaggi@istituto-besta.it

Dear Editor and reviewers,

We were pleased to receive your response letter together with a set of useful comments on our manuscript entitled “Keeping people with dementia or mild cognitive impairment in employment: a literature review on its determinants”, for which we really thank you.

We decided to undertake the revisions and tried to respond to each of the reviewers in the most appropriate way. Changes in the manuscript are highlighted in yellow, so that changes are easily visible to the editors and reviewers.  Responses to reviewers are also reported in the present letter and in the specific form on the submission system.

Thank you for your attention, best regards

Fabiola Silvaggi on behalf of co-authors, Milano – 11/01/2020

Reviewer 2

Comments and Suggestions for Authors

The author provides a comprehensive review including the effects of EOD and MCI on workforce participation.

Following changes need to be addressed:

Comparison of the effects of EOD and MCI on workplace performance between males and females should be discussed in detail. Demographic differences should also be accounted for.

RESPONSE 1: Thank you for this comment, we added in the discussion a paragraph regarding the importance of demographic differences on workplace performance.

'screened' is misspelled in figure 1.

RESPONSE 2: Thank you for noting this involuntary mistake. We corrected it.

data from other diverse sources should be included.

RESPONSE 3: Thank you for this comment, we have added in the introduction other data from other diverse sources with the corresponding references.